# Novel Galectin-3 Roles in Neurogenesis, Inflammation and Neurological Diseases

**DOI:** 10.3390/cells10113047

**Published:** 2021-11-05

**Authors:** Luana C. Soares, Osama Al-Dalahmah, James Hillis, Christopher C. Young, Isaiah Asbed, Masanori Sakaguchi, Eric O’Neill, Francis G. Szele

**Affiliations:** 1Department of Physiology, Anatomy and Genetics, University of Oxford, Sherrington Building, South Parks Road, Oxford OX1 3QX, UK; luana.campossoares@dpag.ox.ac.uk (L.C.S.); isaiah.krikor@gmail.com (I.A.); 2Department of Oncology, University of Oxford, Oxford OX1 3QX, UK; eric.oneill@oncology.ox.ac.uk; 3Irving Medical Center, Columbia University, New York, NY 10032, USA; oa2298@cumc.columbia.edu; 4Massachusets General Hospital, Harvard Medical School, 15 Parkman Street, Boston, MA 02114, USA; jamesmichaelhillis@gmail.com; 5Department of Neurological Surgery, University of Washington, 325 Ninth Avenue, Seattle, WA 98104, USA; chris.uct@gmail.com; 6International Institute for Integrative Sleep Medicine, University of Tsukuba, Tsukuba 305-8575, Japan; masanori.sakaguchi@gmail.com

**Keywords:** galectin, Galectin-3, neurogenesis, subventricular zone, stem cells, inflammation

## Abstract

Galectin-3 (Gal-3) is an evolutionarily conserved and multifunctional protein that drives inflammation in disease. Gal-3’s role in the central nervous system has been less studied than in the immune system. However, recent studies show it exacerbates Alzheimer’s disease and is upregulated in a large variety of brain injuries, while loss of Gal-3 function can diminish symptoms of neurodegenerative diseases such as Alzheimer’s. Several novel molecular pathways for Gal-3 were recently uncovered. It is a natural ligand for TREM2 (triggering receptor expressed on myeloid cells), TLR4 (Toll-like receptor 4), and IR (insulin receptor). Gal-3 regulates a number of pathways including stimulation of bone morphogenetic protein (BMP) signaling and modulating Wnt signalling in a context-dependent manner. Gal-3 typically acts in pathology but is now known to affect subventricular zone (SVZ) neurogenesis and gliogenesis in the healthy brain. Despite its myriad interactors, Gal-3 has surprisingly specific and important functions in regulating SVZ neurogenesis in disease. Gal-1, a similar lectin often co-expressed with Gal-3, also has profound effects on brain pathology and adult neurogenesis. Remarkably, Gal-3’s carbohydrate recognition domain bears structural similarity to the SARS-CoV-2 virus spike protein necessary for cell entry. Gal-3 can be targeted pharmacologically and is a valid target for several diseases involving brain inflammation. The wealth of molecular pathways now known further suggest its modulation could be therapeutically useful.

## 1. Introduction

Galectins are an evolutionarily conserved class of proteins being found in sponges, other invertebrates and mammals [1]. These lectins bind glycosylated proteins and lipids and can be located extracellularly, in the cell membrane and intracellularly. Galectin-3 (Gal-3) is unique amongst the 15 member galectin family (Figure 1A). Gal-3 is “chimeric” by virtue of a c-terminal carbohydrate recognition domain (CRD) and an *N*-terminal non-lectin domain that multimerizes the protein (Figure 1B). In contrast, galectin-1 (Gal-1), a prototypical galectin, has two CRDs and can be anti-inflammatory [2]. Gal-3 binds to *N*-glycans on receptors (Figure 1B), extracellular matrix proteins and pathogens and selectively mediates several key intracellular signaling pathways (Figure 1C). It thereby increases the release of several cytokines and has important functions in inflammation, the immune system and cancer. Gal-3 was previously termed “Mac-2” and was used as a putative marker for activated brain microglia and macrophages. More than just being a marker for them, Gal-3 is secreted by activated macrophages, and is a chemoattractant to them creating a feed-forward cycle [3].

Gal-3 acts as an arsonist at the early stage of inflammation but can act as a fireman at later stages. Gal-3 is upregulated in humans in many CNS diseases associated with inflammation including Alzheimer’s, stroke and hypoxia/ischemia [6,7]. It is also upregulated in rodent models including nerve axotomy, Alzheimer’s disease, multiple sclerosis, stroke and hypoxia/ischemia [8,9,10,11]. We and others find that Gal-3 loss-of-function results in specific histopathological outcomes depending on the disease and its severity. This specificity is likely explained by Gal-3 binding and modulation of divergent glycosylated molecules, including extracellular and intracellular proteins, suggesting it has many functions (Appendix A). Gal-3 was shown to act as a natural paracrine ligand for the Toll-like receptor 4 (TLR4) and to thereby be responsible for inflammation in the lipopolysaccharide (LPS) model of bacterial infection [12]. A number of groups and companies have explored targeting Gal-3 for illnesses including heart disease and fibrosis and several phase II and III clinical trials are in progress. To our knowledge there are no clinical efforts to target Gal-3 in brain disease and inflammation, but given its disease-specific mechanisms of action we believe this a logical next step. It would be important to determine if the Gal-3 inhibitors available so far cross the blood–brain barrier (BBB). If they do, then testing their efficacy in models of brain diseases would be important. If the current inhibitors do not cross the BBB, a new class of Gal-3 inhibitors, capable of crossing the BBB, needs to be developed.

Gal-3 binds extracellular proteins including laminin [13,14], hensin [15], elastin [16], collagen IV [17], tenascin-C, and tenascin-R [18]. On the cell surface, its binding partners include receptors that mediate multiple signaling pathways (Appendix A). Intriguingly, Gal-3 binding can be tissue specific. For example, Gal-3 binds EGFr in non-CNS cell-lines [19] and in solution [20], but we found that Gal-3 does not co-immunoprecipitate with EGFr in mouse SVZ tissue [7,21]. Similarly, while Gal-3 binds VEGFR2 in human umbilical vein endothelial cells [22], this was not the case in the mouse SVZ (Szele lab, unpublished). These discrepancies may be explained by the same protein displaying different tissue-specific glycosylation patterns, a scenario occurring for LRP1, the Prolow-density lipoprotein receptor-related protein 1 [23]. Inside the cell, Gal-3 binds to a different set of proteins. For example, Gal-3 complexes with Rb-CDK4-Cyclin D1 [24], K-Ras [25], Bcl2 [26], β-catenin/Tcf-4 [27,28] and components of the endosomal complex required for transport (ESCRT) [29,30]. For a more extensive list of Gal-3 binding partners, the reader is referred to excellent earlier reviews [31,32,33].

## 2. Galectin-3 Regulates Adult Subventricular Zone Cell Migration

Many more roles for Gal-3 have been uncovered in disease than in homeostasis. Not surprisingly therefore, Gal-3 expression is low in the healthy brain. However, robust Gal-3 expression in the subventricular zone (SVZ) neurogenic stem cell niche in healthy mice is a notable exception. This is of particular interest since SVZ microglia exhibit semi-activated morphologies, even in the absence of injury [34,35]. SVZ microglial morphology ranges from resting ramified to aggressive amoeboid and they express high levels of CD45, and divide more frequently than parenchymal microglia [34,35]. These findings gave clues that Gal-3 may have homeostatic functions in the SVZ niche in healthy brains, and these are described below.

The adult SVZ, which lines the lateral ventricles (LVs) and is the largest neurogenic stem cell niche in the mammalian brain, was originally described in detail by Joseph Altman in the 1960’s, using tritiated thymidine and classic histological methods [36,37]. The SVZ stem cell niche and neurogenesis can be influenced by growth factors, signaling molecules and transcription factors [38,39,40]. SVZ cell types can be distinguished with stage-specific markers, cell-cycle duration, morphological features and transcriptomics [41]. Thus, its cell composition, lineage relationships, molecular mechanisms and responses to disease are fairly well understood. The SVZ contains neural stem cells (NSCs) that proliferate and locally produce transit-amplifying cells (TAPs). TAPs then proliferate and eventually differentiate into neuroblasts, which migrate through the rostral migratory stream (RMS) into the olfactory bulbs (OB) where they mature into specific subtypes of OB interneurons. The adult-born OB neurons are implicated in multiple facets of odor detection, discrimination and memory [42]. Recently there have been a flurry of very useful single cell RNA sequencing (scRNAseq) papers that have indicated even more heterogeneity in the SVZ than previously expected [43,44,45,46,47]. These studies have also found microglia in the SVZ and showed that T cells infiltrate into the aging SVZ in humans [48] and contribute to decreasing neurogenesis in mice [49]. This is of interest since Gal-3 can regulate T cell proliferation, apoptosis and SVZ entry [50].

Neurons are also generated from stem cells in the subgranular zone (SGZ) of the hippocampal dentate gyrus (DG). The SGZ is of particular importance in humans, as it is implicated in memory and affective behaviors, and SGZ neurogenesis is decreased in Alzheimer’s disease (AD) [51,52]. The large majority of mammals that have been investigated exhibit adult neurogenesis. Human SVZ cells are also neurogenic in the first year of life with various groups finding they give rise to olfactory bulb, striatal or cortical neurons [53,54,55]. There is also evidence for human SGZ neurogenesis throughout life with estimates for significant lifetime replacement [52,53]. Whereas some studies cannot find human hippocampal neurogenesis [56], a well-controlled study from the Llorens-Martin lab recently confirmed adult human hippocampal neurogenesis in neurologically healthy individuals [52].

As mentioned above, Gal-3 is not immunohistochemically detectable in most of the brain, but we found it is expressed in the SVZ and the RMS in healthy mice [7,21]. Since Gal-3 is a classic marker of microglia, we expected strong expression in SVZ microglia as they are semi-activated [34]. However, Gal-3 was only minimally expressed by SVZ microglia. Instead it was found in neural cells, namely in ependymal cells (the highest expression), in glial fibrillary acidic protein positive (GFAP+) NSCs, and in some TAPs, but never in neuroblasts. This immunohistochemical expression pattern in the SVZ is strongly maintained in models including stroke [57], mild multiple sclerosis (MS) [58], and severe MS [50]. Transcriptomics analysis with singe cell RNAsequencing indicates a similar SVZ pattern as was found at the protein level, showing LGals3 mRNA is present in astrocyte-like NSCs, TAPs, but not in neuroblasts [59]. That study also found LGals3 transcripts in ependymal cells, smooth muscle cells, microglia and perivascular macrophages [59]. Interrogation of the Allen Brain Atlas shows positive Gal-3 in situ hybridization signals in the SVZ, further indicating Gal-3 is transcribed and translated in the niche rather than diffusing into it. The unique SVZ expression begged the question whether Gal-3 has a function in SVZ homeostasis. We found that loss of Gal-3 in knockout mice did not affect the number of BrdU+ label-retaining NSCs, nor the number of mitotic or apoptotic cells in the SVZ [21]. However, Dcx+ neuroblast chains became disrupted, 2-photon time-lapse microscopy showed reduced speed and straightness of neuroblast migration and overall rates of OB neurogenesis were reduced. These findings suggested that Gal-3 is necessary for maintaining SVZ neuroblast motility [21]. This effect was surprising since Gal-3 is not expressed by the neuroblasts themselves and suggested that Gal-3 was interacting with a cell surface protein. In parallel studies, we found that a subset of neuroblasts continue to express the EGFr and these EGFr+ neuroblasts were slower and less direct in their routes to the OB than EGFr-negative cells [60]. The similarity to the *Gal-3*^−/−^ results on neuroblast migration suggested Gal-3 affects EGFr function and indeed *Gal-3*^−/−^ SVZ cells had higher levels of EGFr phosphorylation [21].

## 3. Galectin-3 Functions in Gliogenesis and Gliomagenesis

### 3.1. Gal-3 Regulates Postnatal Gliogenesis

Since we showed Gal-3 expression and function in the adult SVZ niche, we looked for and found Gal-3 transcripts in the developing embryonic brain (Szele lab unpublished). Future studies could elucidate the role of Gal-3 in embryonic brain development as well as in maternal immune activation paradigms. We next detected Gal-3 in the P5 murine SVZ, in the same cells as adults-NSCs and astrocytes, ependymal cells, and some TAPs and few microglia, but not neuroblasts [7,21]. Supporting this, we showed Gal-3 is expressed in postnatal SVZ neurospheres which are composed of NSCs and progenitors [7,28]. The postnatal SVZ is a major source of forebrain glia [61] and Gal-3 roles in this process were important to study since it had been shown to affect glial biology later in life [62].

Gal-3 activates microglia and inflammation in human pathology and in models of disease such as stroke [63], Huntington’s disease [64], and multiple sclerosis (MS) [50]. However, it was unclear if increased Gal-3 activates microglia in the absence of injury. We therefore overexpressed Gal-3 in the healthy neonatal SVZ and showed that this did not affect microglia activation markers, numbers and morphology [7]. Thus, a model emerged suggesting Gal-3 induces microglial activation only when coupled with tissue damage, such as occurs in stroke, infection, or neurodegeneration. However, this did not rule out other homeostatic effects of Gal-3 in the postnatal brain. Therefore, we studied gliogenesis and neurogenesis in postnatal Gal-3 loss-of-function mice. Floxed Gal-3 conditional knockouts and Gal-3 knockdown reduced gliogenesis but not neurogenesis [7]. In contrast, Gal-3 overexpression in the SVZ increased astrocyte production and maturation in the striatum, while decreasing oligodendrocyte production. Gal-3 overexpression also reduced SVZ proliferation and increased cell-cycle exit [28]. Gal-3 regulates developmental signaling pathways and we asked if Wnt/β-catenin or bone morphogenic protein (BMP) signaling were involved in Gal-3’s postnatal function.

Firstly, Wnt/β-catenin signaling regulates numerous functions in the SVZ and we showed Gal-3 binds β-catenin in SVZ cells [28]. Wnt regulates NSC maintenance [65], symmetric division of NSCs [66] and TAPs [67], as well as oligodendrocytic fate and neuronal differentiation [68,69], suggesting Gal-3/β-catenin binding could have important functions. Inflammation and Wnt signaling blockade mediates age-related NSC quiescence [43] suggesting Gal-3 may play a role in the aging SVZ. We showed that Gal-3 knockdown in the SVZ increased Wnt signaling, whereas overexpression reduced it [28]. Gal-3 may regulate Wnt signaling via binding to β-catenin, sequestering it and inhibiting transcriptional regulation. Whereas Gal-3 reduced Wnt signaling in the SVZ it increases it in pancreatic, breast, colon and tongue cancers [27,70,71,72], as well as in glioblastoma cell-lines (Al-Dalahmah, O–unpublished). The contrast between cancer cell-lines and benign NSCs is of great interest and suggests that tumorigenic transformation alters how Gal-3 regulates Wnt signaling, which remains an open question (please see Section 3.2).

Secondly, we also showed that Gal-3 regulates BMP signaling in the SVZ and binds to the BMPR1A [7]. BMP 2, 4, 6, and 7 ligands and their receptor serine threonine kinases BMPR1A, BMPR1B, and BMPR2, as well as the BMP inhibitor Noggin, are all expressed in the postnatal SVZ niche [73,74]. In the adult SVZ, baseline BMP signaling is required for neurogenesis [73], but increasing BMP levels reduces proliferation and neurogenesis [74,75,76]. BMP signaling also regulates SVZ fate choices by promoting astrocyte generation [77,78,79] and suppressing oligodendrogenesis [73,80,81]. In our study, Gal-3 increased BMP signaling as indicated by increased Smad1/5/8 phosphorylation [7]. Conditional loss of the BMPR1A combined with Gal-3 over expression blocked the increased astrocyte to oligodendrocyte ratio. Interestingly, Gal-3 knockdown did not decrease BMP signaling, suggesting that baseline BMP signaling is maintained via multiple pathways. Together, our results show that Gal-3 coordinates BMP and Wnt signaling in the postnatal SVZ NSCs independent of inflammation, and is a molecular switch for astrocyte genesis in the SVZ.

### 3.2. Gal-3 a Tumorigenesis Marker with Functional Relevance in Gliomagenesis

Cancer is multifactorial with molecular characteristics supporting tumor proliferation and infiltration [82]. Gal-3’s pro-inflammatory role and upregulation of several pro-tumorigenic pathways is linked with cancer aggressiveness. Inflammation, via cytokine secretion and immune cell recruitment, is an attempt to control tumor growth but can instead stimulate tumorigenesis and genomic instability [83,84,85,86,87,88]. Cytokines can activate pathways such as Pi3K/AKT and the Jak/STAT in dedifferentiating tumors and augment cancer stem cells [89,90]. Gal-3 can act at different tumor stages, on multiple cell types and subcellular localizations, a phenomenon not yet completely understood in the context of gliomagenesis. For example, endothelial cell Gal-3 expression is high in low grade glioma but almost absent in glioblastoma, [91,92,93]. In contrast Gal-3 is highly expressed in the parenchyma, in the hypercellular regions surrounding necrosis, supporting its importance for migration or survival of those cells [91]. Gal-3 also modulates the glioma microenvironment via interaction with the glycoprotein chitinase-3-like 1 (CHI3L1), increasing tumor immunosuppression and promoting macrophage M2-polarization [94]. Interestingly, this interaction can be inhibited by Gal-3 binding protein [94].

Gal-3 also has pro-survival roles in tumorigenesis both intra- and extra-cellularly. Gal-3 shares sequence similarity with the NWGR motif of Bcl-2, an antiapoptotic protein [26,95]. Gal-3 also directly binds to Bcl-2 and to other apoptotic proteins involved in apoptosis such as CD95, Nucling and synexin [96,97,98]. Crucially, Gal-3 promotes resistance to anoikis, adhesion-loss induced apoptosis, contributing to tumor dissemination [99,100]. Gal-3 also influences tumor spread through its interaction with integrins, triggering anchorage-independent signaling, facilitating stemness and metastasis [101,102,103].

The uniquely inflammatory status of the SVZ may predispose cancer development via pro-inflammatory regulators such as Gal-3 [35]. Gliomas contacting the SVZ are more aggressive, likely to spread and have a worse prognosis than those not in contact with the stem cell niche [104,105,106,107,108,109,110,111,112,113,114,115]. Gal-3 regulates gliomagenic developmental pathways in the SVZ, enhancing their stem cell characteristics and tumor aggressivity and Gal-3 is a component of the mesenchymal glioblastoma gene signature [116]. Seguin and colleagues have recently shown that Gal-3 regulates micropinocytosis in mesenchymal glioblastoma stem cells, via interaction with Ras related protein 10 (RAB10) and β1 integrin [117].

Cancer-secreted Gal-3 activates Notch signaling impairing differentiation [118,119]. As mentioned, Gal-3 can bind to *N*-glycan residues of tyrosine/kinase receptors EGFR and BMPr1α preventing endocytosis of the former, which ultimately results in upregulation of progenitor genes such as Sox2 [7,19,120]. Notch and EGFR signaling are activated in gliomas contributing to glioma stem cell maintenance [121,122,123,124]. Gal-3 secreted by cancer cells binds to the Notch receptor Jagged-1 and thereby activates angiogenesis [125]. As described above, Gal-3 activates BMP signaling, which controls glioma stem cell quiescence [126,127].

We described above our study showing that Gal-3 binds β-catenin and downregulates Wnt signaling in postnatal SVZ gliogenesis [28]. Wnt pathways are implicated in glioma malignancy and stemness and may be a therapeutic target [128]. Since Gal-3 in the SVZ modulates Wnt signaling opposite to how it is regulated in cancer, SVZ malignant transformation could require a Gal-3 functional switch. In breast cancer, Gal-3 can activate Wnt signaling by mediating β-catenin nuclear localization via direct β-catenin Gal-3 interactions and enhancing Wnt target gene transcription [27,73]. Gal-3 can also indirectly activate Wnt signaling via Akt and GSK3β downregulation in colon [73], pancreatic [72] and tongue cancers [72]. Additionally, Gal-3 can regulate the β-catenin destruction complex as it contains a GSK3β phosphorylation motif and associates with axin [129].

To model early SVZ gliomagenesis, we generated a mouse with conditional IDH1^R132H^ expression in the niche. These IDH1^R132H^ knock-in mice exhibited heightened SVZ proliferation, stem cell expansion and infiltration into adjacent tissue [130]. Gal-3 SVZ expression and microglial activation are heightened in these mice (Figure 2A–D). The enzyme Mgat5 (beta1,6 *N*-acetylglucosaminyltransferase V) adds branched sugars to proteins and galectin binding is proportional to the number of branches [131]. Tumor microenvironments frequently alter glycosylation through abnormal Mgat5 function, which can then alter Gal-3 binding and function [132]. Mgat5 and branched *N*-glycans are related to early gliomagenesis, regulating proliferation and invasion [133,134,135]. These data suggest further Mgat5-mediated roles for Gal-3 in glioma formation and invasion. Gal-3’s actions in promoting brain tumorigenesis and its expression in multiple glioblastoma cell lines (Figure 2E) suggest it could be a good therapeutic target. Interestingly, Gal-3 conferred resistance to traditional treatment with chemotherapy and radiotherapy in glioblastoma [136]. Several inhibitors of Gal-3 have been described and some are in clinical trials for cancer [137,138].

## 4. Galectin-3 Actions in Multiple Sclerosis and Stroke

### 4.1. MS, Gal-3 and the SVZ

Multiple sclerosis (MS) is an autoimmune condition that targets the myelin sheath surrounding neuronal axons [139]. The pathophysiologic process, demyelination, is modeled in animals using both immunogenic and toxic approaches [140]. As expected given the inflammatory nature of demyelination, Gal-3 has an established role in MS. Since, the SVZ generates glia postnatally [61] and Gal-3 regulates SVZ oligodendrogenesis during early postnatal development, it follows that it may do so in regeneration. Interestingly, it has been suggested that SVZ-derived oligodendrocytes may be superior at re-myelination than those derived from oligodendrocyte precursor cells (OPC’s) [141]. Human MS activates the SVZ and its oligodendrocyte production during adult demyelination [142,143,144].

### 4.2. Mgat5-Mediated Sugar Branching Affects MS

SNPs (single nucleotide polymorphisms) in Mgat5 are linked to age of onset and severity in MS [145]. In the mouse MS models, EAE (experimental allergic encephalomyelitis) and TMEV (Theiler’s murine encephalomyelitis virus) susceptible strains exhibit 20-fold less GlcNAc (*N*-acetyl glucosamine) branching. Mgat5 knockouts in these strains exhibit spontaneous demyelination in 80% of mice and more severe EAE [146].

### 4.3. Gal-3 Expression following Demyelination in Humans

Human post-mortem studies in MS find increased Gal-3 mRNA and protein expression (Figure 3A) [50,147,148]. Microglia, macrophages and hypertrophic astrocytes in MS lesions demonstrate Gal-3 expression [147]. Gal-3 is increased in active rather than chronic inactive MS plaques suggesting it actively modulates disease pathology [147]. Gal-3 increases more in primary progressive MS, where MS symptoms steadily worsen from onset, compared to secondary progressive MS, where MS symptoms initially relapse and remit before steadily worsening [148]. There is a particular prominence of Gal-3 expression in the periventricular white matter both within and outside of MS lesions [50]. This is important as MS lesions known as Dawson’s fingers emanate from the lateral ventricles into surrounding regions [149].

### 4.4. Animal Models of MS and Gal-3

Similar to human MS patients, animal demyelination EAE, cuprizone and TMEV models display increased Gal-3 levels [50,150]. Inflammation and Gal-3 increases are especially prominent around the lateral ventricles and SVZ in TMEV [50,151]. In studies in which we used the cuprizone toxin model, Gal-3 expression increased near the SVZ in the white-matter of the corpus callosum [58]. However, after cuprizone treatment Gal-3 expression decreased in the SVZ, a rare occurrence of decreased Gal-3 expression in a disease model. Similar to humans, Gal-3 appears to be localized to parenchymal macrophages, microglia and astrocytes in animal models of the disease, but in the SVZ it is mostly expressed by NSCs, TAPS and ependymal cells [50,58,150,152].

### 4.5. Gal-3 Microglial, Inflammation and Molecular Mechanisms in Demyelination

Gal-3 recruits and activates microglia providing a key role in demyelination. Both the EAE and TMEV models have greater microglial presence in Gal-3 wildtype (*Gal-3^+/+^*) compared to knockout (*Gal-3*^−/−^) mice [50,153]. In the EAE model, the increased microglial presence is associated with increased clinical severity. The cuprizone model has conflicting literature, although it is important to consider the differences in the *Gal-3*^−/−^ mice models used in these studies: not only was the background strain different but, in Hoyos et al. (2014) Gal-3 had an interruption in the CRD by insertion of the neomycin resistance gene at the intro 4-exon 5 junction, while Hillis et al. (2016) eliminated exons 2, 3 and 4 which encode part of both protein domains. Hoyos et al. (2014) report that *Gal-3^+/+^* mice have fewer, yet more activated, microglia than *Gal-3*^−/−^ mice after cuprizone treatment, whereas Hillis et al. (2016) find no difference in the number of hematopoietic cells [58,152]. For microglial activation in the cuprizone model, Gal-3 increases expression of the phagocytic receptor TREM-2b [152]. It has also been proposed that Gal-3 binds to K-Ras-GTP to activate PI3K and phagocytosis [154]. Finally, Gal-3 may be required for matrix metalloproteinase-3 activity, which in turn activates microglia [155,156].

Gal-3 is upstream and required for several pro-inflammatory molecules in MS. Microglia are recruited to the pro-inflammatory milieu of EAE, which Jiang et al. (2009) studied extensively. *Gal-3*^−/−^ lymph nodes produced less interleukin 17 (IL-17), interferon gamma (IFN-γ) and IL-6 in the presence of EAE-inducing myelin oligodendrocyte glycoprotein antigen [157]. In the CNS, *Gal-3*^−/−^ mice but not WT mice had detectable IL-17, IFN-γ, tumor necrosis factor alpha (TNF-α), and inducible nitric oxide synthase (NOS) transcripts. IL-17 promotes blood–brain barrier disruption allowing T(H)17 cell infiltration in MS pathogenesis [139,158]. Gal-3 autoantibodies have separately been suggested to disrupt the blood–brain barrier by binding to Gal-3 on brain microvascular endothelial cells and increasing expression of intercellular adhesion molecule-1 and phospho-nuclear factor-kappa B p65 [159]. In other studies we found greater expression of the chemokines CCL2, CCL5, CCL8 and CXCL10 in the SVZ after TMEV treatment in *Gal-3^+/+^* compared to *Gal-3*^−/−^ mice [50]. We also found that many other chemokine and cytokine expression profiles were decreased in *Gal-3*^−/−^ mice after TMEV (Appendix A) [50]. More recent work upholds the general principle that Gal-3 is pro-inflammatory in the CNS by showing that absence of Gal-3 reduces neuroinflammation in acute peripheral inflammation mouse models [157].

### 4.6. Gal-3 Affects Oligodendrocyte Differentiation

As previously discussed, Gal-3 aids oligodendrocyte differentiation in normal postnatal mice. Separate from its immunologic role in diseases, Gal-3 promotes oligodendrocyte differentiation in demyelination. Gal-3 added to oligodendrocyte cell lines increased differentiation and further increased Gal-3 expression during differentiation [160]. CSF from primary progressive MS patients added to rat oligodendrocyte precursor cells increased Gal-3 expression and increased the number of primary branches from the soma suggesting differentiation [148]. Conversely, in the absence of Gal-3, fewer neurosphere progenitor cells committed to oligodendrocyte fate, and in vivo experiments show decreased myelin integrity in the absence of Gal-3 [160]. *Gal-3^+/+^* but not *Gal-3*^−/−^ mice demonstrated spontaneous remyelination five weeks after cuprizone [152]. Interestingly, *Gal-3*^−/−^ mice demonstrated increased oligodendrocyte precursor cells both at rest and during cuprizone treatment, and had fewer multipolar processes suggesting arrested differentiation [152]. Thus, in aggregate the data suggest Gal-3 is necessary for oligodendrocyte differentiation.

### 4.7. Galectin-3 Functions in Adult Stroke

Stroke leads to a cascade of inflammatory changes and is a leading cause of mortality and disability. Currently, beyond the first few hours where medical thrombolysis and mechanical thrombectomy have yielded impressive results, no treatment is available once the ischemic injury has become established and therefore regenerative approaches are being examined [161]. Gal-3 is garnering clinical interest and serum Gal-3 levels may be useful as a predictor of stroke severity and clinical outcome [162,163]. Given Gal-3’s neuroinflammation roles, it is well-positioned to influence tissue remodeling following ischemic injury. Continued understanding of Gal-3 and its role in post-stroke angiogenesis, neurogenesis and neuroinflammation could contribute to the development of future diagnostic and therapeutic strategies.

To better understand the effect of Gal-3 in stroke, we performed middle cerebral artery occlusion (MCAO) stroke in *Gal-3*^−/−^ knockout mice and compared them to *Gal-3^+/+^* controls. Gal-3 was increased in the area of injury (Figure 3B,C) and deletion of Gal-3 selectively inhibited the stroke-induced increases in endothelial cell proliferation and density in the ischemic penumbra (Figure 3D,E) [10]. Vascular endothelial growth factor (VEGF) and its tyrosine kinase receptors are key regulators of post-stroke endothelial proliferation [164,165]. In *Gal-3*^−/−^ mice, the inhibition of post-stroke angiogenesis was associated with attenuation of the expected upregulation of VEGF, and may be a mechanism for the inhibited endothelial proliferation in *Gal-3*^−/−^ mice after stroke. In contrast to other studies, the reduction in post-stroke angiogenesis in *Gal-3*^−/−^ mice impacted neither stroke size nor functional outcomes [10]. Strategies which increase post-stroke endothelial proliferation appear to reduce stroke size and improve functional outcome, findings which have prompted clinical trials designed to increase angiogenesis. Loss of Gal-3 affected neither inflammation nor proliferation nor neurogenesis in the SVZ. SVZ neuroblasts are diverted from their normal migratory pathway and migrate to the site of ischemic injury. To our surprise, loss of Gal-3 did not affect post-stroke migration of neuroblasts towards the ischemic penumbra. Cytoarchitectural changes such as astrogliosis, endothelial proliferation and loss of ependymal planar cell polarity in the SVZ following ischemic stroke [57] were not affected by *Gal-3*^−/−^ [10]. These results suggest that Gal-3 function in the SVZ can diverge substantially from non-neurogenic parenchymal brain regions. Recent work has continued to indicate that Gal-3 is correlated with and impacts stroke outcomes. Levels of Gal-3 in the serum of patients is associated with severity and progression of ischemic stroke [166]. Treatment with melatonin after ischemic stroke is neuroprotective, reduces levels of Gal-3 and ameliorates hyperactivity and anxiety in rats [167]. However, a recent study showed that treatment with Gal-3 is protective to stroke (MCAO in rats), preventing apoptosis and neurodegeneration [168]. Gal-3 promoted activation of pro-survival pathways such as Akt; and downregulation of pro-apoptotic proteins such as ERK and Caspase-3 [168].

Because of the multiple Gal-3 binding partners and signaling effects, we sought to determine if loss of Gal-3 changes gene expression. We examined striatal protein expression in wild type (WT) and *Gal-3*^−/−^ mice in the presence or absence of MCAO (Appendix A) [10]. Remarkably, Gal-3 loss induced a greater than 10-fold decrease in prolactin expression. *Gal-3*^−/−^ mice also had broad increases in insulin-like growth factor binding proteins IGFBP-1,2,3,9,10, suggesting it regulates insulin signaling. Interestingly, MCAO-induced increases in IGFBP expression were further increased in *Gal-3*^−/−^ compared to WT mice. The same scenario was seen in the expression of thrombospondin-2 and of ADAMTS1 (a disintegrin and metalloproteinase with thrombospondin motifs), suggesting Gal-3 normally limits expression of certain genes involved in angiogenesis.

### 4.8. Gal-3 Functions in Neonatal Hypoxia Ischemia

Gal-3 gene expression is similarly upregulated in a murine model of neonatal hypoxic-ischemic (H/I) injury [169] Unlike our study where deletion of Gal-3 did not affect stroke size and functional outcome, loss of Gal-3 was neuroprotective resulting in reduced ischemic tissue volume [169]. Unexpectedly, this was associated with increased microglial activation and insulin growth factor 1 (IGF-1) expression in *Gal-3*^−/−^ mice. In contrast, targeted ablation of Gal-3+ microglia/macrophages in MCAO was associated with decreased IGF-1 levels but increased apoptosis and stroke size [170]. The same group also demonstrated that deletion of Gal-3 resulted in larger stroke size following MCAO, due to impaired microglial activation and IGF-1 production [170]. In another study in rats, treatment with anti-Gal-3 antibodies attenuated post-stroke endothelial and neural progenitor proliferation in the ischemic striatum and SVZ, respectively [171]. There was no change in stroke size and functional outcome was not reported [171]. In other work, Gal-3 reduced the microglial pro-inflammatory response to LPS and TNFα in vitro [172]. Gal-3 also restored IGF-1 levels after LPS treatment but the same results were not observed in vivo in a model of neonatal hypoxic-ischemia [172].

## 5. Galectin-3 Can Exacerbate Alzheimer’s Disease and Diabetes

### 5.1. Gal-3 Elicits Alzheimer’s Pathology and Symptoms

Amongst molecules implicated in Alzheimer’s disease (AD), Gal-3 has recently emerged as one of the most promising, mechanistically and therapeutically [6,173]. Gal-3 concentrations are increased in the brains, CSF and plasma of humans with AD [174,175,176] and Gal-3 brain injections increase insoluble Aβ levels and toxicity in animals [11,177]. Endogenous Gal-3 is most highly expressed in Aβ plaques [11]. As well, Gal-3 expression increased with age, paralleling Aβ oligomerization [177]. Importantly, SNPs in the Gal-3 gene (LGALS3) are associated with increased risk for AD [11]. Loss of Gal-3 in the severe 5XFAD mouse model of AD decreased microglial Toll-like signaling as well as the AD signature TREM2/DAP12 signaling [11]. Reduced Gal-3 in hemizygote APP/PS1 mice decreased inflammation and enhanced cognition compared to controls [177]. Loss of Gal-3 also ameliorated hippocampus-dependent cognition, suggesting that increased Gal-3 in AD could exacerbate this symptom [11,177]. Another group showed that Gal-3 binding protein (GAL3BP) inhibits β-secretase processing of amyloid precursor protein and thereby reduces Aβ [178].

AD frequently commences in the hippocampus and hippocampal neurogenesis decreases in AD [52]. Stimulating adult hippocampal neurogenesis in rodents improves memory and pattern separation whereas reduced hippocampal neurogenesis decreases memory functions [179,180]. Interestingly, rats undergoing contextual fear conditioning or spatial memory training, which are also adult neurogenesis-dependent behaviors, exhibited reduced hippocampal Gal-3 expression [181]. Reducing Gal-3 increased memory in these tests, whereas overexpression diminished memory, suggesting that Gal-3 normally limits these functions [181]. Whilst all this is intriguing, a major gap in our knowledge is whether Gal-3 directly affects hippocampal neurogenesis, as does Gal-1 (described below).

### 5.2. The Links between AD, Diabetes, Gal-3 and Insulin

A remarkable study showed that in obesity, adipose tissue macrophages secrete Gal-3, which is a chemoattractant to macrophages which then secrete even more Gal-3, in a feed-forward cycle [3]. Gal-3 then binds to the insulin receptor and blocks it, causing insulin resistance [3]. In addition to the insulin receptor, Gal-3 binds to the insulin-like growth factor 1 receptor (IGF1R) [63]. Brain insulin modulates cognition yet how it does so is unclear [182]. These studies are of great interest since the Levison group and others have shown that insulin, IGF-1 and IGF-2 influence various aspects of adult neurogenesis [183]. Whether increased or decreased levels of Gal-3 affect insulin and IGF function in adult SVZ or SGZ neurogenesis is unknown. Type 2 diabetes (T2D) models were recently shown to have decreased hippocampal neurogenesis in both db/db mice (obesity dependent) and IGFr mutant mice (obesity independent) [184]. Hippocampal neurogenesis decreases in human T2D and Gal-3’s role in diabetes may impact neurogenesis [3,185,186]. AD and T2D have significant co-morbidity [187] and given the impact of Gal-3 in AD, it will be important to compare its role in the two diseases. Cerebral microbleeds are common on diabetes melitus and Gal-3 expressing macrophages were associated with abnormal elimination of microvessels at sites of microbleed [188]. Depletion of these macrophages resulted in microvessel repair [188].

## 6. Galectin-3 Relevance in COVID-19 Brain Pathology

### 6.1. Galectins Mediate Viral Infection

A survey of the literature indicates that galectin expression is increased by many viruses via transcriptional regulation or otherwise [189,190]. Galectins as a family have broad roles in regulating viral infections [189,191]. They can act directly on viruses by binding glycosylated viral envelope (env) proteins [190]. Viral glycoproteins can profoundly affect virulence; a single amino acid mutation in the Zika virus env protein caused the 2015 pandemic [192]. Gal-1 is known to bind to the influenza virus and also to decrease flu symptoms [193]. Gal-9, another galectin expressed in the SVZ, binds the human immunodeficiency virus (HIV) virus and HIV increases its expression [190]. The SARS-CoV-2 virus is glycosylated with *N*- and *O*-linked glycan residues located on the receptor binding domain [194,195,196]. The ACE2 receptor is also glycosylated [197] and it is likely that Gal-3 binds to the virus and its receptor. Interestingly, GAL3BP has been shown to bind directly with SARS-CoV-2 and in cell culture models of infection, increasing GAL3BP levels blocked both viral SARS CoV-2 pseudoparticle cell entry and induced cell fusion [198].

### 6.2. Targeting Gal-3 in COVID-19

Two of the greatest risk factors for COVID-19 mortality are obesity and old age. Interestingly, Gal-3 expression is upregulated in both, suggesting it may contribute to the increased inflammation seen in obesity and in old age and that blocking Gal-3 may be a viable therapeutic target [3,11]. Gal-3 inhibitors are being developed for a number of diseases including fibrosis, heart disease and cancer [199,200,201,202,203]. An intriguing suggestion is that they be repurposed for blocking the SARS-CoV-2 virus [204]. This is a logical choice based on Gal-3’s role in inflammation and pathogen response. As mentioned above, Gal-3 is generally pro-inflammatory in the CNS and increases expression of many inflammatory cytokines, for example IL-6 and TNF-α expression via NFKβ [205]. Gal-3 also has well-known roles in infection and pathogen pattern recognition [206,207,208]. Another link is that the Gal-3 CRD shares structural features with coronavirus spike proteins in general [209,210]. The SARS-CoV-2 spike glycoprotein specifically shows remarkable similarity to the Gal-3 CRD. We agree with Caniglia, Velpula and colleagues that it is important to test the ability of these compounds to modulate COVID-19 and also to better understand Gal-3’s role in infection and prognosis of the disease [204].

### 6.3. Does Gal-3 Block Pathogen Entry through the SVZ?

An intriguing question is whether Gal-3 regulates infiltration of pathogens into the SVZ and the brain. SARS-CoV-2 is glycosylated and Gal-3 may intercept it in a proposed network of molecules. A detailed neurological study of CNS pathology reveals that in many cases of COVID-19, encephalopathy is adjacent to or directly impinges on the SVZ (Figure 4A–D) [211]. The SVZ lines the lateral ventricles and along with ependymal cells comprises the cerebrospinal fluid (CSF) brain barrier. However, the barrier is not perfect as SVZ NSC primary cilia extend amongst ependymal cells and contact the CSF in the lateral ventricles. Additionally, we found that loss of Gal-3 causes disruption of ependymal cell motile cilia [21]. We are not aware if increased Gal-3 also causes ciliary problems but if it does, virus could pool in the lateral ventricles. After MCAO stroke, ependymal planar cell polarity was disrupted and we had functional evidence of ciliary dysfunction [57]. Another scenario is that the virus could infect SVZ neuroblasts that would then spread the virus through the brain, since these progenitors frequently move out of the niche and into lesioned areas. The SARS-CoV-2 virus likely has tropism for sialic acid residues [212], and SVZ neuroblasts express polysialylated neural cell adhesion molecule (PSA-NCAM) [213]. In a remarkable instance of viral tropism for the SVZ, we found that the TMEV viral model of MS targets it selectively [50,151]. It is thus important to consider the links between viral entry into the brain via the CSF-brain barrier of lateral ventricles and the expression and function of Gal-3. Even if SARS-CoV-2 does not enter the brain via the lateral ventricles, it likely does through blood vessels disrupted by the virus (Figure 4E). These are frequently surrounded by reactive microglia (Figure 4F) which are likely regulated by Gal-3.

## 7. Galectin-1 Modulates Neurogenesis in the Healthy and Injured Brain

Together with Gal-3, Gal-1 is one of the most studied galectins in the brain. It is also commonly associated with pathological processes such as neuroinflammation. Increasing new evidence supports that Gal-1 and Gal-3 are not redundant molecules, even when expressed in similar contexts, but rather they have defined and specific actions. Therefore, it is relevant to consider the role of Gal-1 in the healthy and diseased brain. A more detailed comparison of Gal-1 and Gal-3 in neuroinflammation can be found in [214].

Amongst the 15 galectins, Gal-3 is chimeric and therefore in a class by itself, whereas Gal-1, the first galectin to be discovered, belongs to the “prototypcal galectins”, a subfamily including Gal 1, 2, 5, 7, 10, 11, 13, 14 and 15 (Figure 1A). Gal-1 is a 14.5 kDa protein expressed in most mammalian organs, however, its expression pattern in the adult brain is limited to specific subtypes of astrocytes and neurons [215]. Gal-1 functions in at least three structural states: as a monomer, a dimer lectin and non-lectin, when disulfide bonds are formed [216]. When it acts as a lectin, it binds to glycans consisting of lactosamine sequences. Gal-1 functions both inside and outside cells, and is secreted using a non-canonical trans-golgi pathway.

### 7.1. Gal-1 Functions in the SVZ

It has been suggested that Gal-1 is expressed in the adult SVZ NSCs [215]. The Gal-1 expressing cells are sparsely distributed amongst GFAP+ SVZ astrocytes, but not in nearby striatal non-neurogenic astrocytes. Adult SVZ NSCs are slowly dividing, therefore, the adult NSC population is enriched with BrdU label-retaining astrocytes [217]. Interestingly, Gal-1 is expressed in a subset of these BrdU-label retaining cells, but not in the other SVZ cells, suggesting selective expression of Gal-1 in adult SVZ NSCs [215,218].

We have compared the lectin and non-lectin forms of Gal-1 by infusing their recombinant proteins into mouse LVs and found that only the lectin-form of Gal-1 increased the number of SVZ progenitor cells [215]. Infusion of Gal-1 and anti-Gal-1 neutralizing antibodies increased and decreased the number of BrdU-label retaining cells, respectively, suggesting that the number of the adult SVZ NSCs is positively regulated by the lectin form of Gal-1. In addition, infusion of Gal-1 protein or knocking down the *LGals-1* gene consistently increased and decreased the number of transit amplifying progenitors (TAP) cells, respectively [215]. However, it remained unclear whether this effect is due to direct binding of Gal-1 to TAP cells or through regulation of SVZ neural stem cells, which produce TAPS.

To examine which SVZ cell types the lectin form of Gal-1 binds in the brain, a biotinylated lectin form of Gal-1 was applied to adult brain sections [215]. Gal-1 binding was found on TAPS, neuroblasts, and sometimes on stem cells. On the other hand, mass spectrometric analysis of SVZ tissues using affinity columns showed that Gal-1 binds the β1 subunit of integrin (β1-Integrin). Infusion of β1-Integrin neutralizing antibody blocked Gal-1-induced increases in the number of BrdU label-retaining SVZ cells. β1-integrin plays critical roles in cell adhesion and is expressed by GFAP+ stem cells and DCX+ neuroblasts. Indeed, in vitro analysis using SVZ neurospheres revealed that Gal-1 regulates adhesion in concert with β1-integrin [219]. Finally, we analyzed downstream regulators of β1-integrin on SVZ adhesion by quantifying Akt phosphorylation. Akt Ser473 phosphorylation in a subset of SVZ astrocytes was significantly reduced in *Gal-1*^−/−^ mice, suggesting Gal-1 is needed for cell-adhesion mediated SVZ signaling [219]. Taken together, these results suggest that Gal-1 regulates SVZ NSC proliferation through regulating cell adhesion via binding carbohydrate structures on β1-Integrin.

### 7.2. Gal-1 Function in the Diseased Brain

Gal-1 function in the injured brain was examined using mouse models of brain ischemia [220] and spinal cord injury [221]. Gal-1 expression increased markedly in SVZ and peri-lesion striatal astrocytes in the ischemia model. As discussed above, brain ischemia increases SVZ neurogenesis and ectopic neuroblast migration to damaged brain regions. Interestingly, both neurogenesis and ectopic neuroblast migration were blocked by anti-Gal-1 neutralizing antibodies but enhanced by infusion of the lectin-form of Gal-1, but not by the non-lectin form [220]. Importantly, the Gal-1 lectin improved recovery from ischemia-induced functional deficits. We have also shown Gal-1 ameliorates functional recovery from spinal cord injury models in rodents and in the common marmoset [222]. Gal-1 expression is ubiquitous outside the CNS so it may be difficult to specifically pharmacologically target Gal-1 in the CNS. Nevertheless, it may be possible to target specific carbohydrate structures to which Gal-1 binds on NSCs or reactive astrocytes.

### 7.3. Gal-1 Regulates Hippocampal Neurogenesis

Immunohistochemical and morphological studies revealed that Gal-1 is expressed in a subset of DG NSCs and their immediate descendants, transit amplifying progenitor cells [218]. Histological analysis of *Gal-1*^−/−^ mutant mice injected with BrdU revealed that Gal-1 inhibits early stem and progenitor SGZ proliferation, however other work suggests it increases kainite induced SGZ neurogenesis [223]. Indeed, the number of adult DG granular neurons was increased in *Gal-1*^−/−^ mice. The mutant mice showed deficits in two types of hippocampal memory tests with normal level of anxiety [224]. The memory deficits could be explained by DG adult-neurogenesis functions in forgetting [225], although it is unknown whether the increased adult DG neurons in the mutant mice is the direct cause or not for the observed behavioral phenotypes. Further analysis by conditionally removing Gal-1 from NSCs may provide further insights.

## 8. Conclusions

The data discussed above show the powerful and broad effects that Gal-3, and to a lesser extent, Gal-1 has in brain health and disease. Remarkably, despite multiple binding partners, dozens of signaling targets and myriad intra- and extra-cellular functions, Gal-3 has very specific context-dependent effects. For example, it regulates neuroblast migration in the adult SVZ and in the postnatal niche it only regulates gliogenesis. Another example is that Gal-3 is necessary for neovascularization in the penumbra of stroke but not in the SVZ. How can this be? Theoretically Gal-3 loss or overexpression should be promiscuous, causing a complex set of cellular effects. Gal-3’s specialized functions are all the more unexpected since its loss results in an array of downstream molecular events both in homeostasis and in disease. One mechanism which could explain divergent cellular mechanisms of action in the SVZ is redistribution of its cellular pattern of expression. However, irrespective of the model of disease, Gal-3 cell type expression remains remarkably stable in the SVZ neurogenic niche. As for the parenchyma, where Gal-3 expression is usually undetectable with immunohistochemistry, it typically increases in microglia and in astrocytes in pathological contexts.

Whilst SVZ Gal-3 function in health and disease has been studied, its function in the SGZ of the hippocampal dentate gyrus remains obscure. Whether Gal-3 increases in the DG after injury is poorly explored. We speculate that Gal-3 immune barrier functions in the SVZ contribute to its expression. Gal-3 and Gal-1 have overlapping functions in adult neurogenesis but with some distinctions. They share several binding partners and general functions. It may well be that their co-expression in the niche helps to balance proliferation versus migration in adult homeostasis.

We have learned much about Gal-3 function in neurogenesis and brain pathology, but several important questions remain. It is unclear how Gal-3 expression in the SVZ influences the resting semi-inflamed status of the adult SVZ. We showed that modulation of Gal-3 levels in healthy neonates does not seem to modify inflammation [7], however it is not clear if this is the case in adults. It may be that Gal-3 expression in the SVZ is upregulated by inflammatory cytokines in the niche thus creating a feedforward cycle. In general more needs to be learned about how Gal-3 is regulated. Although Gal-3 is typically upregulated in brain injury and disease, we found that it was decreased in the SVZ in the cuprizone model of MS. What were the upstream mechanisms that caused this? Another unanswered question is how Gal-3 regulates signaling pathways within different cells of the SVZ. We found that both BMP and Wnt signaling were regulated by Gal-3 in the same SVZ cells. How is this coordinated? The new floxed Gal-3 mouse from EUCOMM, [7] will help dissect some of these issues. Use of scRNAseq to compare transcriptomics of Gal-3 loss-of-function to baseline will help dissect the role of Gal-3 at different stages of the lineage.

## Figures and Tables

**Figure 1 cells-10-03047-f001:**
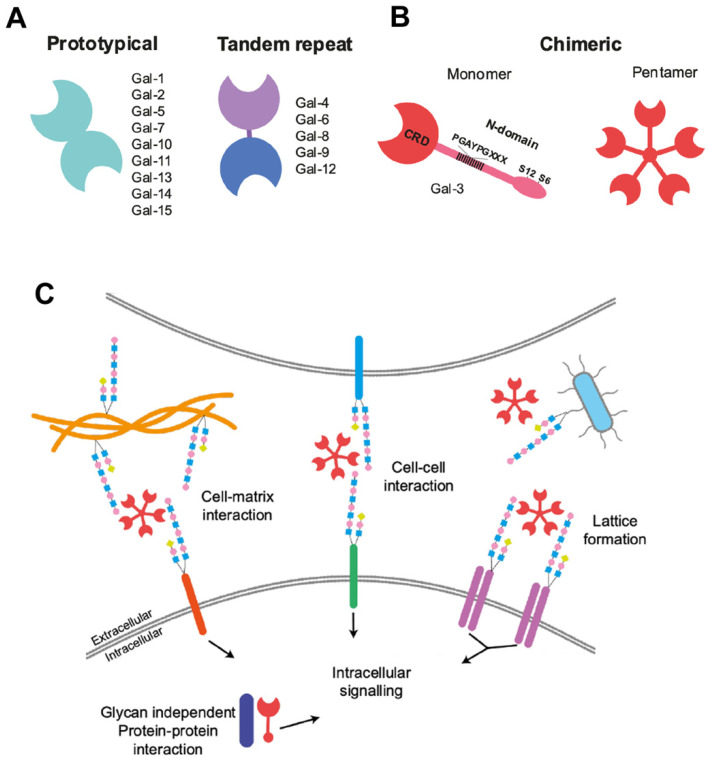
Schematics of Galectin-3 structure and function (**A**) Galectin subtypes and homology. Adapted from [4], (**B**) *N*-glycan binding partners of Gal-3. Adapted from [5], (**C**) Galectin-3 interactions. Adapted from [4], with permission.

**Figure 2 cells-10-03047-f002:**
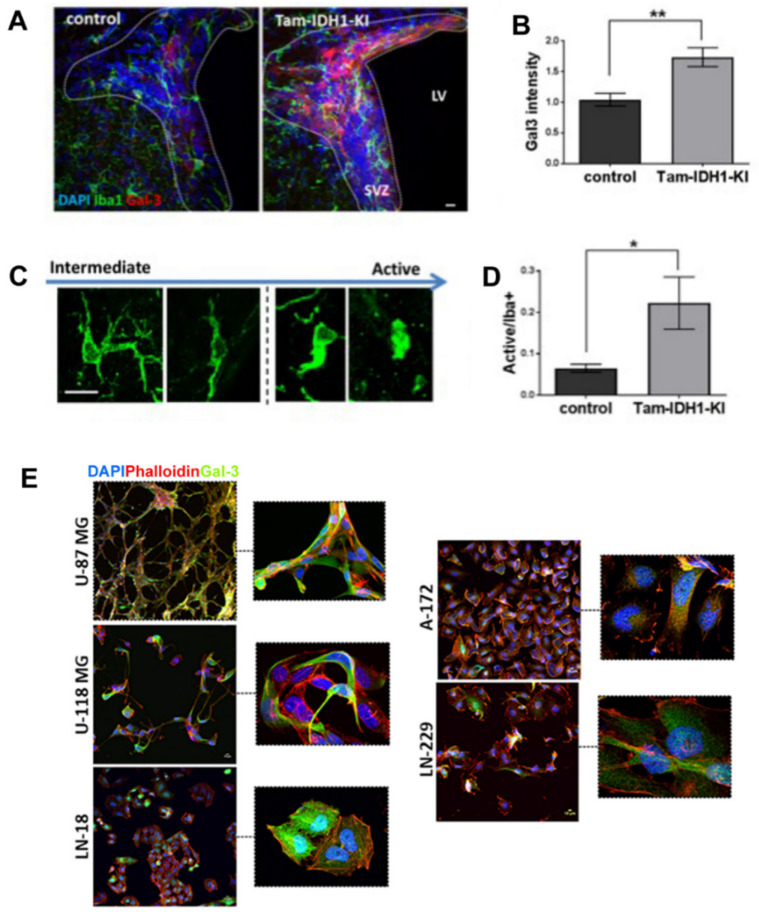
Galectin-3 expression and microglia in an SVZ cancer model and in cancer cells. (**A**) Gal-3 expression (red) and microglial Iba1 expression (green) are increased in the SVZ of the IDH1^R132H^ model of gliomagenesis as described in Bardella et al., 2016. Scale bar is 5 microns. (**B**) Quantification of A showing significantly increased Gal-3 in the glioma model SVZ. (**C**) Typical Iba1+ microglial morphologies in the SVZ ranging from intermediate to activated. (**D**) Quantification of C showing that in the IDH1 model there are significantly more active microglia in the SVZ. (**E**) Gal-3 expression (green) is found in five glioblastoma cell lines. Phalloidin (red) used to detect actin. ((**A**–**E**): Szele, Tomlinson, Bardella and O’Neil labs, unpublished data). * *p* < 0.05, ** *p* < 0.01.

**Figure 3 cells-10-03047-f003:**
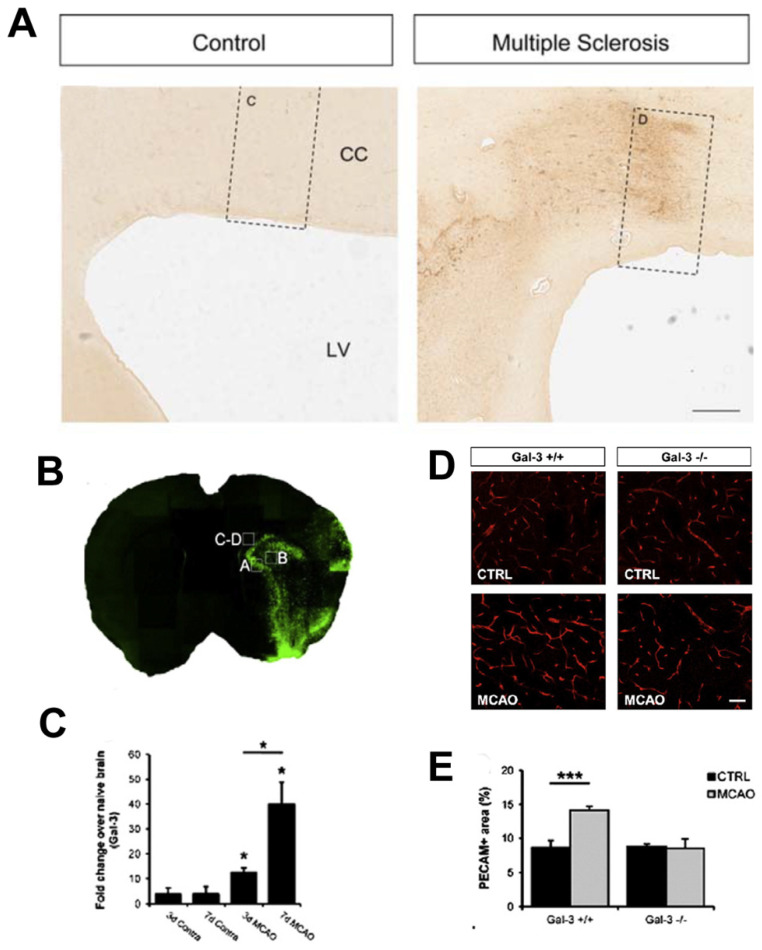
Galectin-3 is upregulated in multiple sclerosis and controls angiogenesis after stroke. (**A**) Gal-3 immunohistochemistry (brown) in human brain sections shows that compared to a healthy control, Gal-3 is increased near the lateral ventricle (LV) in an MS patient. (Adapted from [50]. (**B**) Gal-3 immunofluorescence (green) is increased in the striatum and cerebral cortex of a mouse middle cerebral artery occlusion (MCAO) stroke model. (**C**) Quantification of B showing significantly increased Gal-3 at days 3 and 7 post-stroke. (**D**) Platelet endothelial cell adhesion molecule (PECAM) blood vessel immunofluorescence (red) in the striatum of WT and *Gal-3*^−/−^ mice, with or without MCAO. (**E**) Quantification of D showing increased blood vessels after MCAO in controls but not in *Gal-3*^−/−^ mice. ((**B**–**E**) Adapted from [10]), with permission. * *p* < 0.05, *** *p* < 0.001.

**Figure 4 cells-10-03047-f004:**
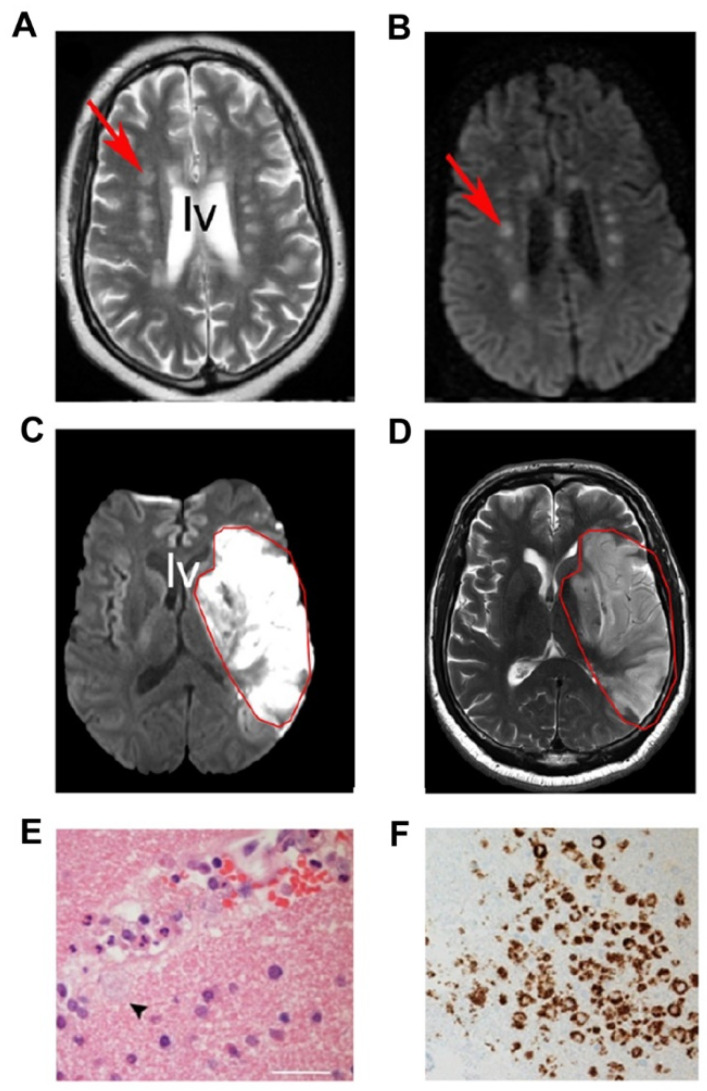
CNS pathology in COVID-19 victims. (**A**,**B**) MRI showing small foci of injuries (arrows) near the lateral ventricle (LV) and SVZ. (**C**,**D**) Large lesion (outlined in red) near the lateral ventricles. (**E**) A small blood vessel surrounded by immune cells that invaded the brain. Note macrophage extending into brain (small arrowhead). (**F**) CD68 immunohistochemistry showing macrophages around small vessels. (Adapted from Paterson et al., 2020), with permission.

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
