# Peer review of "Novel Galectin-3 Roles in Neurogenesis, Inflammation and Neurological Diseases"

_cells, 2021, doi:10.3390/cells10113047_

Round 1
Reviewer 1 Report
Galectin-3 plays an important role in physiological functions of the nervous system, but it is also implicated in variety of neurological disorders. Authors discussed it’s detail role and covered role of Gal-3 in inflammation and brain disease. The current manuscript in clear and written in nicely, but similar article in MDPI discussing about its role is overlapping, author should revisit the below article and compare the present study, so that novel and recent finding will be further added to the current understanding on Gel-3 role in over all biology.
Biomolecules. 2020 May; 10(5): 798.Published online 2020 May 21. doi: 10.3390/biom10050798
Galectin-3: Roles in Neurodevelopment, Neuroinflammation, and Behavior
I will suggest editing the images in Figure 2 E (looks very small) and it’s hard to pint out author point, similarly in Figure 3 B and D.
Author Response
Reviewer 1:
Galectin-3 plays an important role in physiological functions of the nervous system, but it is also implicated in variety of neurological disorders. Authors discussed it’s detail role and covered role of Gal-3 in inflammation and brain disease. The current manuscript in clear and written in nicely, but similar article in MDPI discussing about its role is overlapping, author should revisit the below article and compare the present study, so that novel and recent finding will be further added to the current understanding on Gel-3 role in over all biology.
Biomolecules. 2020 May; 10(5): 798.Published online 2020 May 21. doi: 10.3390/biom10050798
Galectin-3: Roles in Neurodevelopment, Neuroinflammation, and Behavior
Thank you for the positive comments and suggestions. We have added this review article, as well as another. We refer to them as follows. “For a more extensive review of Gal-3 binding partners, the reader is referred to excellent earlier reviews [1-3].”
We have also added, to Table 1, a few more Gal-3 binding partners that have been recently published.
We have also added several new references in various sections of the review.
I will suggest editing the images in Figure 2 E (looks very small) and it’s hard to pint out author point, similarly in Figure 3 B and D.
Thanks for the suggestions, we have enlargened Fig. 2E and Fig. 3B,D.
Reviewer 2 Report
This review is a generally well-written and comprehensive overview of Galectin-3 functions in adult neurogenesis and across a variety of neurological conditions. It should provide a useful reference for both those new to the field and for those with more experience. In its current state, the manuscript has some minor inconsistencies in writing between sections that I would hope to see improved, especially between the Abstract/Introduction and the more detailed subsections that follow. I would also hope to see some more detail on the unpublished and/or reanalyzed data represented in the Figures and Tables; a list of suggested minor edits is below. Overall, however, the review represents an excellent synthesis of expertise on galectins; with some minor edits, it will be a useful resource for several fields of study.
Minor comments and edits:
- The title could be more specific to help the review appear more easily in searches. It should spell out “Galectin-3” rather than use an abbreviation, and it should use a more specific term than “brain disease,” for instance “neurological diseases” or “neurodegenerative diseases,” to make a distinction from neurodevelopmental disorders.
- Language: While the writing is generally very clear throughout the manuscript, the Abstract and Introduction require some editing to use more precise scientific language. For instance, in the Abstract Line 16, “ancient and fascinating” could be rewritten to “evolutionarily conserved and multifunctional”. Similar editing for more precise scientific language is needed in Lines 53-54.
- Abstract Line 21: Do the authors mean “TREM2 (triggering receptor expressed on myeloid cells)” rather than “TRIM2 (tripartite motif-containing protein 2)”? I am not aware of literature showing that Gal-3 and TRIM2 are binding partners, nor are any such studies discussed in the text of the manuscript.
- Abstract Line 28: Instead of “matches the SARS-CoV-2 virus spike protein”, I would suggest more nuanced language such as “bears structural similarity to the SARS-CoV-2 spike protein.”
- Lines 147-148: Can the authors refer to published single-cell or sorted cell RNA-Seq data to provide further evidence on which cell types transcribe Gal-3 under basal conditions?
- Lines 106-107: A reference should be included for Altman’s studies.
- Figure 2: If novel unpublished data are presented in the review, then a detailed Methods section (including ethics approval for animal research) needs to be included in the manuscript. Panel C also needs more descriptive information in the Figure Legend: where in the brain are the images taken, scale bar dimensions, etc.
- Lines 375-76: The wording here is confusing: “only Gal-3-/- mice but WT had no detectable…”
- Table 2: The labelling “B6+/+” vs “B6-/-“ can be confusing and does not match the figure legend with “Bb-/-“, or the main text that uses “Gal-3-/-“. Can the table use the same nomenclature as the text?
- Table 2: While the information included here is useful, it would be even better to see which of the genes show significant (adjusted p<0.05 or other cutoff) changes between Gal-3+/+ and Gal-3-/- mice. P-values should be included in addition to fold changes.
- Table 3: The legend does not specify what is indicated by the blue text (fold above control Gal-3+/+, presumably). P-values should also be included here, along with more detail on the methods used.
- Lines 485-487: A reference is needed for this statement.
- Lines 561-562: A reference is needed.
- Lines 582-584: A reference is needed for Gal-1’s expression patterns among cell types.
Author Response
Reviewer 2:
This review is a generally well-written and comprehensive overview of Galectin-3 functions in adult neurogenesis and across a variety of neurological conditions. It should provide a useful reference for both those new to the field and for those with more experience. In its current state, the manuscript has some minor inconsistencies in writing between sections that I would hope to see improved, especially between the Abstract/Introduction and the more detailed subsections that follow. I would also hope to see some more detail on the unpublished and/or reanalyzed data represented in the Figures and Tables; a list of suggested minor edits is below. Overall, however, the review represents an excellent synthesis of expertise on galectins; with some minor edits, it will be a useful resource for several fields of study.
Thank you for the positive comments and suggestions.
Minor comments and edits:
- The title could be more specific to help the review appear more easily in searches. It should spell out “Galectin-3” rather than use an abbreviation, and it should use a more specific term than “brain disease,” for instance “neurological diseases” or “neurodegenerative diseases,” to make a distinction from neurodevelopmental disorders.
Done – thanks for the good ideas, we have taken them on board and re-titled the article as: “Novel galectin-3 roles in neurogenesis, inflammation and neurological diseases.”
- Language: While the writing is generally very clear throughout the manuscript, the Abstract and Introduction require some editing to use more precise scientific language. For instance, in the Abstract Line 16, “ancient and fascinating” could be rewritten to “evolutionarily conserved and multifunctional”. Similar editing for more precise scientific language is needed in Lines 53-54.
We agree and the changes have been made. Reviewer 3 also suggested we change lines 53-54 and we have integrated their suggested re-wording.
- Abstract Line 21: Do the authors mean “TREM2 (triggering receptor expressed on myeloid cells)” rather than “TRIM2 (tripartite motif-containing protein 2)”? I am not aware of literature showing that Gal-3 and TRIM2 are binding partners, nor are any such studies discussed in the text of the manuscript.
Many thanks for catching this mistake – we have corrected it now.
- Abstract Line 28: Instead of “matches the SARS-CoV-2 virus spike protein”, I would suggest more nuanced language such as “bears structural similarity to the SARS-CoV-2 spike protein.”
This is an excellent suggestion and we have changed the wording.
- Lines 147-148: Can the authors refer to published single-cell or sorted cell RNA-Seq data to provide further evidence on which cell types transcribe Gal-3 under basal conditions?
This is also an excellent suggestion and we have included the following: “Transcriptomics analysis with singe cell RNAsequencing indicates a similar SVZ pattern as was found at the protein level, showing LGals3 mRNA is present in astrocyte-like stem cells, transit amplifying progenitor cells, but not in neuroblasts [4]. That study also found LGals3 transcripts in ependymal cells, smooth muscle cells, microglia and perivascular macrophages [4].”
- Lines 106-107: A reference should be included for Altman’s studies.
Excellent point, thank you, we agree and have inserted two of his references.
- Figure 2: If novel unpublished data are presented in the review, then a detailed Methods section (including ethics approval for animal research) needs to be included in the manuscript. Panel C also needs more descriptive information in the Figure Legend: where in the brain are the images taken, scale bar dimensions, etc.
We have generated a new Supplemental Materials section describing Fig. 2, detailing the experimental methods, and including ethics approval for animal research. We have also added more detail on scale bars and brain location in the figure legend.
- Lines 375-76: The wording here is confusing: “only Gal-3-/- mice but WT had no detectable…”
We agree this is confusing and have changed it to: “In the CNS, Gal-3-/- mice but not WT mice had detectable IL-17, IFN-γ, tumor necrosis factor alpha (TNF-α), and inducible nitric oxide synthase (NOS) transcripts.”
- Table 2: The labelling “B6+/+” vs “B6-/-“ can be confusing and does not match the figure legend with “Bb-/-“, or the main text that uses “Gal-3-/-“. Can the table use the same nomenclature as the text?
We agree and have harmonised the nomenclature in the Table with the text: (Gal-3+/+ and Gal-3-/).
- Table 2: While the information included here is useful, it would be even better to see which of the genes show significant (adjusted p<0.05 or other cutoff) changes between Gal-3+/+ and Gal-3-/- mice. P-values should be included in addition to fold changes.
That is a good idea but these were technical replicates not biological replicates. Therefore, we placed the table in the supplementary section in our James et al, 2016 Glia paper, and feel that we would need biological replicates and/or qPCR confirmation to carry out statistics.
- Table 3: The legend does not specify what is indicated by the blue text (fold above control Gal-3+/+, presumably). P-values should also be included here, along with more detail on the methods used.
Yes, that is correct, the values in blue are the precent of non-lesioned (naïve) WT Gal-3 mice, as we indicate in the Table 3. We have now expanded Table 3’s legend to give more methodological detail. “Protein expression Proteome Profiler Mouse Angiogenesis Array was used. Protein expression as percentage of positive control spot is shown in black font. Protein expression as percentage of naïve Gal-3+/+ wildtype control is shown in blue font. Brain homogenates 1 week after MCAO were prepared and quantified as for Western blots to obtain protein samples. Protein samples from individual animals in each group were pooled (n = 3). Samples were incubated for 1 hour with biotinylated antibodies to permit subsequent detection by Strepavidin detection antibodies. Nitrocellulose membranes were spotted with positive control antibodies and a series of “capture” antibodies in duplicate. Adapted from Young et al., 2014.” Similar to point number 10 we did not have sufficient technical or biological replicates to do statistics. We had place this data in the Supplementary section of our Young et al., 2014 Exp Neurol Paper and would need Western blot confirmation to carry out statistics.
- Lines 485-487: A reference is needed for this statement.
Done. “Stimulating adult hippocampal neurogenesis in rodents improves memory and pattern separation whereas reduced hippocampal neurogenesis decreases memory functions [5, 6].”
- Lines 561-562: A reference is needed.
Done and we have toned down the strength of the comment based on new data. “The SARS-CoV-2 virus likely has tropism for sialic acid residues [7], and SVZ neuroblasts express polysialylated neural cell adhesion molecule (PSA-NCAM) [8].”
- Lines 582-584: A reference is needed for Gal-1’s expression patterns among cell types.
Done. “Gal-1 is a 14.5kDa protein expressed in most mammalian organs, however, its expression pattern in the adult brain is limited to specific subtypes of astrocytes and neurons (Sakaguchi et al., 2006).”
Reviewer 3 Report
The present manuscript is well written and comprehensive.
However, this reviewer would like to raise some concerns.
- The title of the manuscript is not appropriate if the authors want to include Galectin-1 and SARS-CoV-2 in the manuscript. The authors should either change the title or remove the content about galectin-1 and SARS-CoV-2. Regarding galectin-1, if the authors want to keep discussion on galectin-1, besides changing the title, they should compare the role of galectin-3 with galectin-1 throughout the text and not as a subtopic of discussion. Regarding SARS-CoV-2, there is not enough evidence, if any, on the role of galectin-3 in SARS-CoV-2 infection and the discussion is merely speculative.
- Line 52-53: Please review this metaphor. "Gal-3 is like a fireman who is an arsonist. It waits around, not doing much, until there is inflammation and then it springs into action by adding to the inflammation.". The role of galectin-3 as an arsonist is comprehensive, still, the role of galectin-3 as a fireman cannot be conceived in this metaphor. Maybe it would be worth mentioning that gal-3 acts as an arsonist at the early stage of inflammation but as a fireman at later stages.
- "To our knowledge there are no clinical efforts to target Gal-3 in brain disease and inflammation, but given its disease-specific mechanisms of action we believe this a logical next step.". Regarding this sentence, the authors could easily check if the galectin-3 inhibitors available so far cross the blood brain barrier (BBB). If so, to test for brain diseases is a logical next step. If not, a new class of gal-3 inhibitors, capable of crossing the BBB, needs to be developed.
- I could not find this article on the reference list (Table 1, line 4): Dos Santos et al., 2017.
- "For a more extensive review of Gal-3 binding partners, the reader is referred to an earlier excellent review (Dumic et al., 2006)." There must be more updated information on this regard.
- Please check the first time you write MS and add initials meaning.
- "Several inhibitors of Gal-3 have been described and some are in clinical trials for cancer (Stegmayr et al., 2019; Wdowiak et al., 2018). ". Do they cross the BBB?
- Gal-3 knockout mice, Gal-3-/- mice. Need to standardize how to refer to galectin-3 knockout mice.
- "The cuprizone model has conflicting literature: Hoyos et al (2014) report that Gal-3+/+ mice have fewer, yet more activated, microglia than Gal-3-/- mice after cuprizone treatment, whereas Hillis et al (2016) find no difference in the number of hematopoietic cells (Hillis et al., 2016; Hoyos et al., 2014)." - Regarding the conflict literature, it might be worth mentioning that galectin-3 knockout mice used in the two studies differ not only in the background (Sv129 vs C57) but in the way the galectin-3 gene was knocked out.
Author Response
Reviewer 3:
The present manuscript is well written and comprehensive.
Thank you for this positive comment.
However, this reviewer would like to raise some concerns.
- The title of the manuscript is not appropriate if the authors want to include Galectin-1 and SARS-CoV-2 in the manuscript. The authors should either change the title or remove the content about galectin-1 and SARS-CoV-2. Regarding galectin-1, if the authors want to keep discussion on galectin-1, besides changing the title, they should compare the role of galectin-3 with galectin-1 throughout the text and not as a subtopic of discussion. Regarding SARS-CoV-2, there is not enough evidence, if any, on the role of galectin-3 in SARS-CoV-2 infection and the discussion is merely speculative.
This is an important point. We included Gal-1 as it is another galectin which has been shown to affect adult neurogenesis. Overall there is much less known about Gal-1 function in homeostasis and post-pathology neurogenesis compared to Gal-3. Nevertheless, we have added new information at the beginning of the Gal-1 section to compare the two galectin functions. The large majority of the review is on Gal-3 and thus we feel that it would be misleading to include Gal-1 in the title.
With regards to SARS-CoV-2, the links with Gal-3 are significant and we believe it should be included in the review. We provide several non-speculative facts that strengthen this link. For example, we discuss evidence for Gal-3 binding to viral proteins. We believe that carefully chosen speculation is appropriate, especially in a review article. Mere facts frequently do not move science forward without speculation, please see recent essay in Nature pointing this out [9].
- Line 52-53: Please review this metaphor. "Gal-3 is like a fireman who is an arsonist. It waits around, not doing much, until there is inflammation and then it springs into action by adding to the inflammation.". The role of galectin-3 as an arsonist is comprehensive, still, the role of galectin-3 as a fireman cannot be conceived in this metaphor. Maybe it would be worth mentioning that gal-3 acts as an arsonist at the early stage of inflammation but as a fireman at later stages.
Done - thanks for the suggestion, the metaphor is now more precise. “Gal-3 acts as an arsonist at the early stage of inflammation but can act as a fireman at later stages.”
- "To our knowledge there are no clinical efforts to target Gal-3 in brain disease and inflammation, but given its disease-specific mechanisms of action we believe this a logical next step.". Regarding this sentence, the authors could easily check if the galectin-3 inhibitors available so far cross the blood brain barrier (BBB). If so, to test for brain diseases is a logical next step. If not, a new class of gal-3 inhibitors, capable of crossing the BBB, needs to be developed.
Yes, we could do that but it would require some time and resource. We have added your suggestions to the text. “It would be important to determine if the Gal-3 inhibitors available so far cross the blood brain barrier (BBB). If they do, then testing their efficacy in models of brain diseases is a logical next step. If the current inhibitors do not cross the BBB, a new class of Gal-3 inhibitors, capable of crossing the BBB, needs to be developed.”
- I could not find this article on the reference list (Table 1, line 4): Dos Santos et al., 2017.
It is [10]. We have now included it as well as the following sentence. “Gal-3 secreted by cancer cells binds to the Notch receptor Jagged-1 and thereby activates angiogenesis [10].”
- "For a more extensive review of Gal-3 binding partners, the reader is referred to an earlier excellent review (Dumic et al., 2006)." There must be more updated information on this regard.
Good point – we have added two as follows. “For a more extensive review of Gal-3 binding partners, the reader is referred to excellent earlier reviews [1-3].”
- Please check the first time you write MS and add initials meaning.
Thanks – done.
- "Several inhibitors of Gal-3 have been described and some are in clinical trials for cancer (Stegmayr et al., 2019; Wdowiak et al., 2018). ". Do they cross the BBB?
This is a good question and we do not know the answer. Please see response to your question 3.
- Gal-3 knockout mice, Gal-3-/- mice. Need to standardize how to refer to galectin-3 knockout mice.
Thanks – done. We now refer to them systematically as Gal-3-/-.
- "The cuprizone model has conflicting literature: Hoyos et al (2014) report that Gal-3+/+ mice have fewer, yet more activated, microglia than Gal-3-/- mice after cuprizone treatment, whereas Hillis et al (2016) find no difference in the number of hematopoietic cells (Hillis et al., 2016; Hoyos et al., 2014)." - Regarding the conflict literature, it might be worth mentioning that galectin-3 knockout mice used in the two studies differ not only in the background (Sv129 vs C57) but in the way the galectin-3 gene was knocked out.
Thanks, this is a good point. We now state the following. “The cuprizone model has conflicting literature, although it is important to consider the differences in the Gal-3-/- mice models used in these studies: not only was the background strain different but, in Hoyos et al (2014) Gal-3 had an interruption in the CRD by insertion of the neomycin resistance gene at the intro 4 - exon 5 junction, while Hillis et al (2016) eliminated exons 2, 3 and 4 which encode part of both protein domains.”
References
- Dumic, J., S. Dabelic, and M. Flogel, Galectin-3: an open-ended story. Biochim Biophys Acta, 2006. 1760(4): p. 616-35.
- Srejovic, I., et al., Galectin-3: Roles in Neurodevelopment, Neuroinflammation, and Behavior. Biomolecules, 2020. 10(5).
- Johannes, L., R. Jacob, and H. Leffler, Galectins at a glance. J Cell Sci, 2018. 131(9).
- Zywitza, V., et al., Single-Cell Transcriptomics Characterizes Cell Types in the Subventricular Zone and Uncovers Molecular Defects Impairing Adult Neurogenesis. Cell Rep, 2018. 25(9): p. 2457-2469 e8.
- Anacker, C. and R. Hen, Adult hippocampal neurogenesis and cognitive flexibility - linking memory and mood. Nat Rev Neurosci, 2017. 18(6): p. 335-346.
- Miller, S.M. and A. Sahay, Functions of adult-born neurons in hippocampal memory interference and indexing. Nat Neurosci, 2019.
- Sun, X.L., The Role of Cell Surface Sialic Acids for SARS-CoV-2 Infection. Glycobiology, 2021.
- Szele, F.G. and M.F. Chesselet, Cortical lesions induce an increase in cell number and PSA-NCAM expression in the subventricular zone of adult rats. J Comp Neurol, 1996. 368(3): p. 439-54.
- Nurse, P., Biology must generate ideas as well as data. Nature, 2021. 597(7876): p. 305.
- Dos Santos, S.N., et al., Galectin-3 acts as an angiogenic switch to induce tumor angiogenesis via Jagged-1/Notch activation. Oncotarget, 2017. 8(30): p. 49484-49501.